# Establishment of Residual Methods for Matrine in Quinoa Plants and Soil and the Effect on Soil Bacterial Community and Composition

**DOI:** 10.3390/foods12061337

**Published:** 2023-03-21

**Authors:** Xiangjuan Hui, Hongyu Chen, Shuo Shen, Hui Zhi, Wei Li

**Affiliations:** 1Academy of Agriculture and Forestry, Qinghai University, Xining 810016, China; 2College of Agriculture and Animal Husbandry, Qinghai University, Xining 810016, China; 3State Key Laboratory of Plateau Ecology and Agriculture, Qinghai University, Xining 810016, China; 4Key Laboratory of Integrated Management of Agricultural Pest in Qinghai Province, Xining 810016, China

**Keywords:** QuEChERS, liquid chromatography triple quadrupole tandem mass spectrometry (LC-MS/MS), matrine, quinoa (*Chenopodium quinoa* Willd.) plant, soil, pesticide residues, bacterial community and composition

## Abstract

A method was developed for the determination of matrine residues in quinoa (*Chenopodium quinoa* Willd.) plants and soil by liquid chromatography triple quadrupole tandem mass spectrometry (LC-MS/MS) with QuEChERS clean-up. Matrine from soil, quinoa roots, stems, leaves and seeds was extracted with 25% ammonia, 20 mL acetonitrile/methanol, salted with sodium chloride (NaCl) and purified with anhydrous magnesium sulfate (MgSO_4_), N-propyl ethylenediamine (PSA) and graphitized carbon black (GCB). Then a chromatographic separation was performed on a Shim-pack XR-ODS II (75 mm × 2.0 mm, i. d., 2.2 µm) column with a gradient elution of 5 mmol/L ammonium formate-methanol as the mobile phase and monitored in multiple reaction monitoring modes (MRM) in electrospray positive ionization mode. The results showed that in the range of 0.005~1 mg/L, the linear correlation coefficients of matrine in the five matrices were all above 0.999. The LOQs for soil, quinoa roots, stems, leaves and seeds were 0.005, 0.005, 0.01, 0.01 and 0.005 mg/kg, respectively. The mean recoveries ranged from 74.42% to 98.37%, with RSDs of 1.25–6.84% at the three concentration addition levels. The average intra-day and inter-day recoveries were 73.92–92.36% and 78.56–90.18%, respectively, with RSDs below 8.72% and 9.43%. The recoveries and reproducibility of the method were superior. The method was used to determine the actual samples, which indicated that the half-lives of matrine in quinoa seeds, leaves, stems and soil were 1.28–1.32, 1.03–1.21, 0.81–0.92 and 0.93–0.97 d. It has a half-life below 30 d, which is an easily dissipated pesticide. The method is simple, sensitive, accurate, reliable and applicable to a wide range of applications, and it can achieve the rapid multi-residue determination of matrine to a certain extent. Next Generation Sequencing was used to explore the effects of exposure to high and low doses of matrine on soil bacterial communities and the composition of the three soils in the Qinghai Province (Haixi, Haidong and Haibei). The results showed that the number of ASVs increased significantly after treatment with matrine at an effective dose of 0.1 mg/kg than after treatment with matrine at an effective dose of 5.0 mg/kg. Similarly, bacterial abundance was higher after 0.1 mg/kg of matrine treatment than after 5.0 mg/kg of matrine treatment. The inhibitory effect on some bacterial flora was enhanced with an increase in matrine application, while the inhibitory effect on bacterial flora was weakened with time. Applying a certain dose of matrine e changed the relative abundance of the dominant bacterial genera of the soil bacteria.

## 1. Introduction

Quinoa (*Chenopodium quinoa* Willd.), a genus of quinoa in the Amaranthaceae family, is also known as quinoa grain, southern quinoa, and quinoa. The high altitudes of the Andes in South America are its origin, with nearly 7000 years of plant cultivation. It was one of the main foods of the ancient Inca indigenous people and was called the “mother of grains” [1]. It is mainly grown experimentally in Africa, Europe and Asia. Quinoa prefers cold and dry but mild relative humidity, highland and alpine areas. The soil needs to be well drained with pH levels from six to nine. The climatic, geographic, and soil conditions of the Qaidam Basin in the Qinghai Province almost replicate the growth and cultivation environment of the Andean highlands of South America, being the best suitable area for quinoa cultivation in the Tibetan Plateau [2]. Quinoa contains all the essential amino acids and is rich in high-quality protein and many trace elements, which makes it a “whole food” [3]. In recent years, people have gradually become aware of the nutritional value of quinoa, with higher demands on its quality and flavor, leading to an increase in the area under cultivation. During its growth, various pests and diseases have emerged, with an increasing trend year by year. At present, the reported diseases of quinoa include downy mildew, leaf spot, black stem, grey mold, root rot, etc., among which downy mildew and leaf spot are the most severe damage to quinoa cultivation in China; insect pests include cutworm, mole cricket, scarab beetles and *Plutella xylostella* [1,4]. To meet the constant demand for quinoa quality and yield, producers are forced to use pesticides unscientifically, blindly and in large quantities, resulting in resistance and tolerance of pests and weeds, making the efficacy of pesticides gradually decrease or even disappear. At the same time, long-term reliance on chemical pesticide control, resulting in surface pollution, water pollution and other problems, is increasingly presented.

Matrine is a quinolizidine alkaloid isolated from *Sophora japonica*. Molecular formula: C_15_H_24_N_2_O; relative molecular mass: 248.37; CAS No.: 519-02-8; relative density 1.16; melting point 77 °C; boiling point 396.7 °C; the appearance of the original drug is a white powder. It can be soluble in water, benzene, chloroform, methanol, ethanol, and slightly soluble in petroleum ether [5,6,7]. The matrine molecule contains two nitrogen atoms, N16 in the amide state, which is weakly basic, and N1 knotted in the ring as a tertiary amine. Its steric configuration facilitates the acceptance of protons, thus effectively blocking the effects arising from the space-site blocking effects; hence it has a strong basicity [8,9]. The chemical structural formula is illustrated in Figure 1. Domestic matrine formulations include 0.6% aqueous matrine, 0.8% lactone aqueous matrine, 1% matrine solution, 1.1% matrine solution, 1.1% matrine powder, etc. Pesticide products with matrine as the active ingredient have been widely used on various crops such as fruits, vegetables, tea and tobacco [10,11]. The agent has the characteristics of low toxicity, spectrum and safety, with excellent insecticidal and antibacterial effects. It is effective against diseases such as gray mold of grapes, late blight of potatoes, black star disease of pear trees, downy mildew of zucchini, mycorrhizal disease of rape and virus disease of tobacco, as well as pests such as cabbage moth, aphid, tea looper, tea black poison moth, root-knot nematode and chard moth [12,13,14,15]. More than 90 companies in China have registered 115 single or compounded formulations with matrine as the active ingredient. Although matrine is a plant-derived insecticide, it is generally considered to be environmentally compatible. Matrine has a half-life of 6.7–21.9 d in cucumber and kale soils [16] and a half-life of 7.64 d in tobacco [17]. It struggles to move deep into the soil, is concentrated in the shallow 0–10 cm layer of soil and is easily degraded in pond water and river water [18]. However, excessive amounts of matrine can be toxic to humans and livestock; it may paralyse the respiratory system and cause nephrotoxicity. To provide a scientific basis for rational drug use, its safety needs to be evaluated by establishing a stable and convenient assay with high sensitivity [19,20,21]. The “GB2763-2021, National food safety standard: maximum residue limits for pesticides in food” specifies a maximum residue level of 5 mg/kg for matrine on kale, cucumber and pear and 1 mg/kg on orange, tangerine and mandarin. However, no national test standard detection methods were given for testing the residues of matrine in these substrates. Fewer reports involve the detection of residues in soil, and the detection of residues in quinoa has not been reported yet [22]. Currently, there are liquid mass spectrometry [15,23], gas chromatography [16,24], high-performance capillary electrophoresis [25] and thin layer scanning [26] methods for the qualitative and quantitative determination of matrine. Most of the existing domestic literature reports and assays are about the determination of matrine bases in traditional Chinese medicine and other aspects [27,28,29,30,31]; however, the quinoa matrix contains an increased number of special components such as pigments, organic acids and phenols compared to traditional Chinese medicine. It is easy to interfere with the confirmation of the target to varying degrees. Microorganisms are important indicators of soil ecosystem health [32,33] and can characterize changes in soil quality more rapidly than physical and chemical indicators in the soil. The bacterial community structure of the soil is involved in energy flow and elemental cycling in ecosystems [34], such as rhizobacteria and nitrifying bacteria [35]. It affects the fertility status of the soil and alters the microbial diversity of the soil and the carbon source utilization capacity of the microbial community [36,37].

Given this, this study was conducted to establish a method for the rapid determination of matrine residues in quinoa plants and soil by QuEChERS-LC-MS/MS. It will provide technical support for the safety of quinoa and give methodological references for the determination of other kinds of pesticides in the soil environment. Next Generation Sequencing of bacteria in the soil to determine changes in bacterial communities provides a theoretical basis for the rational application of matrine.

## 2. Materials and Methods

### 2.1. Chemicals and Reagents

The chemicals and reagents used are as follows. Methanol (purity, 100%), acetonitrile (purity, 100%), acetic acid (purity, 100%), formic acid (purity, ≥98%), ammonium formate (purity, ≥99%), ammonium acetate (purity, ≥99%, chromatographic grade, Merck, Darmstadt, Germany); anhydrous magnesium sulfate, sodium chloride and ammonia (analytical grade, Sinopharm Chemical Reagent Co., Ltd., Shanghai, China); ethylenediamine-*N*-propyl silane (PSA), graphitized carbon black (GCB), Florisil and octadecyl bonded silica gel (C_18_) (Angela Technologies, Tianjin, China). Matrine standard (purity, 95.2%, Pesticide Quality Supervision and Inspection Center, Shenyang Research Institute of Chemical Industry, Shenyang, China); 0.6% matrine aqueous (Inner Mongolia Qingyuanbao Biotechnology Co., Ltd., Bayannur, China).

### 2.2. Field Experiments

Dissipation experiments: field experiments were designed according to the requirements stated in the Guidelines for Pesticide Residue Tests and the Standard Operating Procedures for Pesticide Registration Residue Field Tests [38,39]. Each plot was 15 m^2^ and applied at a rate of 720.36 g a.i./hm^2^ matrine, with 0.6% aqueous active ingredient and with 1, 2 and 3 applications each. Samples were collected at 2 h, 1 d, 3 d, 5 d, 7 d and 14 d after application. Each treatment was set up in three replicates with a protective isolation zone between treatments and blank control.

Residue experiments: two application rates of 360.18 g a.i./hm^2^ (recommended rate) and 720.36 g a.i./hm^2^ (twice the recommended rate) were set; experiments were conducted in accordance with the Guidelines for Testing Pesticide Residues in Agricultural Crops (NY/T788-2018) [38]. Each treatment was applied 1, 2 and 3 times, with 3 replicates for each treatment. The sampling intervals were 3 d, 7 d and 14 d from the last application. There was a blank control and a protection zone between treatments.

Sample collection and preparation: roots, stems, leaves, and quinoa seeds were collected randomly per plot, their impurities were removed, and they were cut. The samples were shrunken to over 1 kg and packed in closed plastic bags. Sampling labels were glued and stored in a refrigerator at −20 ± 2 °C. The soil was selected from a 0–10 cm tillage layer and filtered through a 40 mesh sieve to remove impurities. Soil samples were stored as in quinoa plants. The basic physicochemical properties of the test soils were determined by conventional analytical methods (Table 1).

### 2.3. Equipment, Mass Spectrometry and Chromatographic Conditions

Equipment: a liquid chromatography-triple quadrupole tandem mass spectrometer (LC-MS/MS 8040 Shimadzu, Kyoto, Japan), a high-capacity high-speed centrifuge (TGL-50, Changzhou Shen Guang Instrument Co., Ltd., Changzhou, China), an ultrasonic cleaner (SB-3200DTD, Nanjing Emmanuel Instruments Co., Ltd., Nanjing, China), a vortex mixer (Vortex Genie 2; Scientific Industries, Inc., Bohemia, NY, USA), and an electronic analytical balance (SQP, Sartorius Scientific Instruments Co., Beijing, China).

Mass spectrometry conditions: the molecular mass of matrine is 249, which has a strong ionization efficiency in ESI positive ion mode; the atomization gas flow rate was 3.0 L/min; the drying gas flow rate was 15 L/min; the DL temperature was 250 °C; the heating block temperature was 450 °C. According to the results of the precursor ion scan and the product ion scan of matrine, the precursor ion *m*/*z* = 249 and product ions *m*/*z* = 148, 97.95 and 55 were monitored. The quantitative ion was *m*/*z* = 148, which was used for the qualitative and quantitative analysis of matrine. Other mass spectrometry parameters are listed in Table 2.

Chromatographic conditions: the analytical column was a Shim-pack XR-ODSⅡ(75 mm × 2.0 mm, i. d., 2.2 µm); the flow rate was 0.3 mL/min; the injection volume was 1 μL; the column temperature was 40 °C; the mobile phase included A as methanol and B as a solution containing 5 mmol ammonium formate; the mobile phase gradient elution was: 0~1.5 min, 10% A; 1.5~2 min, 10~90% A; 2~6.5 min, 90% A; 6.5~7.5 min, 90~10% A; 7.5~8 min, 10% A.

### 2.4. Sample Preparation and Clean-Up Procedure

Soil: weigh 10.00 g of soil (accurate to 0.01 g) in a 50 mL stoppered centrifuge tube, add 2 mL of 25% ammonia, vortex well and let stand for 10 min, then add 20 mL of acetonitrile, vortex and shake for 3 min to mix well. Add 1 g NaCl and 4 g anhydrous magnesium sulfate, vortex and shake for 1 min; centrifuge at 5000 r/min for 5 min. Take 1.5 mL of supernatant in a 2 mL centrifuge tube with 100 mg PSA and 100 mg anhydrous magnesium sulfate, vortex for 1 min, and centrifuge at 12,000 r/min for 2 min. Filter the supernatant through a 0.22 μm membrane and transfer it to a liquid chromatography glass vial for measurement.

Quinoa roots, stems and seeds: weigh 2.00 g (to 0.01 g) of quinoa root, stem and seed samples in a 50 mL stoppered centrifuge tube, add 3 mL of 25% ammonia, vortex well, and let stand for 10 min. Add 20 mL of acetonitrile, vortex and shake for 3 min, mixing well. Add 1 g NaCl and 4 g anhydrous magnesium sulfate, vortex and shake for 1 min; centrifuge at 5000 r/min for 5 min. Add 1.5 mL of the root and stem supernatants into a 2 mL centrifuge tube pre-spiked with 100 mg PSA, 20 mg GCB and 100 mg anhydrous magnesium sulfate. Take 1.5 mL of the seed supernatant and add to a 2 mL centrifuge tube pre-spiked with 70 mg PSA, 20 mg GCB and 100 mg anhydrous magnesium sulfate, vortex for 1 min and centrifuge at 12,000 r/min for 2 min. Filter the supernatant through a 0.22 μm membrane and transfer it to a liquid chromatography glass vial for measurement.

Quinoa leaves: use 20 mL of methanol as an extractant, 150 mg of PSA, 20 mg of GCB and 100 mg of anhydrous magnesium sulfate as a purifying agent. Other pretreatment conditions were the same for quinoa roots, stems and seeds.

### 2.5. Standards

Weigh 10 mg of matrine standard precisely, dissolve it with methanol and fix the volume in a 100 mL brown volumetric flask. Dissolve it by ultrasonication and prepare the standard stock solution with a concentration of about 100 mg/L, then keep it in a cooler at −18 °C away from light. Accurately measure 1 mL of matrine standard stock solution, dilute it step by step with a blank sample extract, and prepare matrix-matched standard solutions with mass concentrations of 0.005, 0.01, 0.05, 0.1, 0.5 and 1.0 mg/L. Matrine was measured under the above conditions. The standard curve was plotted with the mass concentration of the sample as the horizontal coordinate and the corresponding peak area as the vertical coordinate.

### 2.6. PHI and Risk Assessment

PHI can be defined as the number of days between the last application of pesticide to the crop and harvest [40,41]. It is determined based on the maximum residue limit (MRL) values and degradation curves. PHI can be calculated with the help of the following formula [41]:PHI = [Ln (initial deposit) − Ln (MRL)]/slope of the regression equation

The pesticide dietary exposure assessment methodology was assessed using the NEDI model with the following equations [42]:NEDI = STMR × Fbw (1)
RI = NEDIADI × 100% (2)
where F is the domestic dietary consumption per capita (kg/d) and bw (body weight) is the body weight per capita (kg); all body weights in China are calculated at 63 kg. STMR is the median residue test value (mg/kg); ADI (Acceptable daily intake) is the allowable daily intake of pesticides (mg/kg bw); RI is the risk index, where an RI < 100% indicates that the dietary risk is in an acceptable range, while the opposite indicates that the risk is unacceptable, and the higher the value, the higher the risk.

### 2.7. Next Generation Sequencing to Study—The Effect of Matrine on Different Soil Bacterial Community

Soils from three locations in the Qinghai Province (Haixi, Haibei and Haidong) were collected for this experiment, and ultra-pure water was added to bring the water content of the soil to 24% (equivalent to 60% of the field water holding capacity). Moreover, 0, 1 and 50 µg of matrine were added to give a concentration of 0 (control), 0.1 (low dose) and 5.0 mg/kg (high dose) to the soil. They were vortexed for 5 min, mixed, and then preincubated in a constant temperature incubator at 25 ± 2 °C protected from light. The soil water content was maintained by regular water replenishment. Samples were taken at 3 d and 10 d and stored in a cold room at −18 °C for measurement (Table 3).

Three replicates were set up for each treatment group. Fifty-four soil samples (including three replicates) were selected from three locations in the Qinghai Province: Xiangrid in Haixi, Jintan Township in Haibei and Ledu in Haidong. The 16SrDNA Next Generation Sequencing analysis of bacteria was performed. Total genomic DNA was extracted from the samples using the CTAB/SDS method, and the quality of the total extracted DNA was checked by 1% agarose gel electrophoresis. PCR was performed using the Phusion^®^ High-Fidelity PCR Master Mix from New England Biolabs, Ipswich, MS, USA. The PCR reaction conditions are as follows: 98 °C (10 s), 50 °C (30 s) and 72 °C (30 s) for 30 cycles; 72 °C (5 min). The PCR products were examined by electrophoresis using a 2% concentration agarose gel. Next Generation Sequencing was performed using NovaSeq 6000 (Illumina, San Diego, CA, USA). Sequencing was performed by Beijing NovoSeq Technology Co. (Beijing, China).

The Clean Tags were compared with the reference database using Vsearch (Version 2.15.0) to detect the chimera sequences, and then the chimera sequences were removed to obtain the Effective Tags [43]. The spliced sequences were further processed using the QIIME2 (Version QIIME2-202006) analysis software to obtain the initial ASV; ASVs with an abundance of less than 5 were filtered out [44]. In order to analyze the diversity, richness and uniformity of the communities in the sample, alpha diversity was calculated from 7 indices in QIIME2, including Observed_otus, Chao1, Shannon, Simpson, Dominance, Good’s coverage and Pielou_e. To find out the significantly different species at each taxonomic level (Phylum, Class, Order, Family, Genus, Species), the R software (Version 3.5.3) was used for MetaStat and *t*-test analysis.

## 3. Results and Analysis

### 3.1. Optimization of Chromatographic Conditions

The relative abundance of the target ions to be measured in the organic solvent methanol and acetonitrile was examined. Methanol is a proton-like solvent that forms hydrogen bonds and has improved selectivity for separating compounds with acids and bases. Acetonitrile is a polar molecule. Its carbon-nitrogen triple bond contains unbonded electrons and can bind to compounds containing empty orbitals. The results showed that the response intensity and peak area of matrine in methanol were better than those in acetonitrile, as indicated in Figure 2. Therefore, methanol was used as the organic phase of the mobile phase in the experimental optimization. Given the attainment of a good peak shape, response value and a high signal-to-noise ratio, aqueous solutions of 0.01%, 0.1% and 0.2% formic acid and acetic acid, aqueous solutions of 5 mmol, 10 mmol and 15 mmol ammonium formate, an aqueous solution of 5 mmol ammonium acetate, an aqueous solution of 0.1% formic acid-5 mmol ammonium formate and an aqueous solution of 0.1% acetic acid-5 mmol ammonium acetate were investigated as mobile phases, respectively. The results demonstrate that the addition of acid to the aqueous solution in the mobile phase resulted in an early peak time, low response value and broad peak shape; the addition of 5 mmol ammonium formate or ammonium acetate to the aqueous phase could adjust the pH value of the solution, improve the ionization efficiency and the peak area response value of the mass spectrometry, and improve the peak shape of the target compounds, and also result in a high signal-to-noise ratio. However, the relative abundance of ions was higher when 5 mmol ammonium formate was used. Then the ammonium formate dosages of 5 mmol, 10 mmol and 15 mmol were optimized. There was no significant difference in the effects of the three ammonium formate dosages on the ion response intensity and peak shape, etc. Therefore, 5 mmol ammonium formate was selected as the aqueous phase by experimental investigation.

### 3.2. Optimization of Mass Spectrometry Conditions

Matrine is a weakly basic compound containing nitrogen in its chemical structure, and due to the easily additive protons with positive charges, its suitable for detection in ESI+ mode [45,46]. Based on the molecular structure characteristics and chemical ionization properties of the compounds, the signal differences of the compounds in positive and negative ion modes of ESI sources were examined, respectively. In this research, the compounds’ parent ions were first obtained by a preliminary mass spectrometry scan (Q1 MS) using a needle pump with continuous injection in positive ion mode. Then it was subjected to a sub-ion full scan. Two fragment ions with strong response signals were selected, the one with a higher response value being the quantitative ion and the other being the qualitative ion. The parent ion and these two fragment ions were then formed into a detection ion pair, and the declustering voltage (DP) and collision energy (CE) were optimized in MRM mode [47]. The optimization results are shown in Table 4. After the optimized daughter ion pair, cleavage voltage, and collision energy, the compound had a more robust mass spectrometry signal response. The stability and reproducibility of the mass spectrometric determination were enhanced to meet the quantitative limit requirements.

### 3.3. Optimization of Extractants

#### 3.3.1. Selection of Extractants for Different Matrices

Alkaloids can be divided into lipophilic alkaloids and water-soluble alkaloids according to their solubility properties. Matrine has tertiary amine bases with a small molecular weight, which are soluble in both lipophilic organic solvents such as benzene, ether, and trichloromethane, etc., and hydrophilic organic solvents such as methanol and acetonitrile, as well as in aqueous alkali solutions [48,49]. When lipophilic organic solvents are used, extraction, centrifugation, rotary evaporation and concentration processes are required. The operation procedures are complicated, so to shorten the extraction time and save organic solvents, the effects of two hydrophilic extractants, methanol and acetonitrile, on the recovery of matrine in different matrices were screened. The results showed (Figure 3A) that the average recoveries of matrine in soil, root, stem, leaf and seed were 10.48%, 66.10%, 67.36%, 43.36%, 54.79% and 23.57%, 74.26%, 71.21%, 40.07%, 69.75% when methanol and acetonitrile were used as extractants, respectively. Thus, methanol was selected as the best extractant of matrine in leaves and acetonitrile as the best extractant of matrine in soil, roots, stems and seeds.

#### 3.3.2. Optimization of the Dosage of Different Matrix Extractants

The effects of three dosage volumes (10 mL, 20 mL, 40 mL) of methanol or acetonitrile on the average recovery effect of matrine in different matrices were compared. The results showed that (Figure 3B): the average recoveries of matrine in soil were 17.72%, 33.48% and 18.36% when the extraction solvent volumes were 10, 20 and 40 mL, respectively; and the average recoveries were higher when 20 mL acetonitrile was selected as the soil extractant. The average recoveries of matrine in roots were 58.24%, 67.78% and 68.56%, respectively. The average recoveries of matrine in stems were 61.38%, 66.14%, and 66.21%, respectively. The average recoveries of matrine in leaves were 38.91%, 43.80% and 45.09%, respectively. The average recoveries of matrine in the seeds were 51.67%, 60.43% and 60.51%, respectively. The average recovery of matrine in the five matrices increased with the extractant dosage, and there was no significant difference in the average recovery of matrine at 20 mL and 40 mL of extractant, respectively. Hence, to reduce the number and amount of organic solvents as well as the risk to humans, 20 mL acetonitrile was selected as the soil, root, root stem and seed extractant and 20 mL methanol as the leaf extractant.

#### 3.3.3. Optimization of Ammonia Dosage for Different Substrates

The chemical structure of matrine contains amino (-NH_2_), which is weakly basic, and it is known from a literature survey that matrine is better extracted under weakly basic conditions [50]. Therefore, experiments were conducted to investigate the extraction recovery of matrine at 20 mL acetonitrile + 1, 2, 3 and 5 mL ammonia, respectively, using soil, quinoa roots, stems, and seeds as substrates, and quinoa leaves as substrates at 20 mL methanol + 1, 2, 3 and 5 mL ammonia, respectively. The results showed that (Figure 3C) the average recoveries of matrine in the four extracts were: 72.74%, 97.86%, 94.36% and 90.26%, 72.72%, 76.68%, 83.56% and 83.31%, 68.40%, 72.26%, 75.44% and 75.95% for soil, quinoa root, stem, leaf and seed substrates, respectively; 75.96%, 74.45%, 71.25%, and 70.36%, 76.64%, 77.65%, 78.63%, and 77.80%. Improving the recovery rate, reducing the amount of ammonia and the environmental risk were considered. Consequently, 20 mL acetonitrile + 2 mL ammonia water was selected for soil, 20 mL acetonitrile + 3 mL ammonia water for quinoa roots, stems and seeds, and 20 mL methanol + 1 mL ammonia water for quinoa leaves as the best extraction solvent.

#### 3.3.4. Selection of Sorbent

The QuEChERS method is fast, efficient, time-saving and simple to operate, eliminating the need for complex purification processes. It reduces the pretreatment time of organic solvents, thus reducing the harm to the human body. Commonly used purification agents for the Qu ECh ERS method are ethylenediamine-*N*-propyl silane (PSA), octadecyl bonded silica gel (C_18_), Florisil, graphitized carbon black (GCB), etc.

Soil purification: it is complex to remove impurities from the soil matrix that interfere with the detection of the target compounds. The effects of different doses of PSA, C_18_, Florisil and GCB on the recovery of matrine were compared. As shown in Figure 4A–D, PSA had no significant adsorption effect on matrine, with recoveries mostly ranging from 80–100%, while GCB, Florisil and C_18_ had strong adsorptions on matrine. The adsorption of impurities by 30 mg PSA was unsatisfactory, and the recovery was low. The recovery increased when the PSA content was greater than 50 mg. When the PSA content was more significant than 100 mg, the recovery was unchanged with an increase in the PSA dose. The recoveries of C_18_, Florisil and GCB were below 75% when they were used as purifying agents, for which the recoveries of matrine gradually decreased with an increase in purifying agent dosage. GCB produced a more substantial adsorption effect on the compounds. When comparing the four purifying agents, PSA recovered relatively quickly and also removed most of the impurities in the soil, with less interference of spurious peaks and low pigment content in the soil. Moreover, PSA (100 mg) was used to purify the soil samples.

Purification of the quinoa plant: quinoa is rich in vitamins, polyphenols, flavonoids, saponins and phytosterols. It also has high protein, 83% unsaturated fatty acids in its fat content and high pigment content [51,52]. An examination of the effect of different doses (30 mg, 50 mg, 70 mg, 100 mg, 150 mg) of PSA, C_18_, Florisil and GCB (20 mg, 40 mg, 60 mg, 80 mg, and 100 mg) on the recovery of matrine from quinoa plants was required. As shown in Figure 4A, the recoveries of quinoa roots were above 80% when different doses of PSA were used. At 100 mg PSA, the recovery was 96.54%, and at 150 mg, the recovery showed a decreasing trend. The recoveries of quinoa stems were between 65–80%; with increasing PSA doses, the recoveries of matrine gradually increased. When the PSA dose was more than 100 mg, there was no significant increasing trend in the recovery. The recovery of quinoa leaves was 75–85%. The recoveries were above 80% at 70 mg PSA. The dose of purgative increased, but there was no significant increasing trend in the recoveries. For quinoa seeds, the recoveries were in the range of 80–95%. With the same effect on soil, quinoa root, stem and leaf substrates, all of which gradually increased with an increasing PSA dose matrine recovery, the highest recovery was achieved at 150 mg PSA. The recoveries of quinoa roots, stems, leaves and seeds were between 60–75%, 45–65%, 70–75% and 60–80% when different doses of C_18_ were used, depending on the matrix. As shown in Figure 4B, the recoveries of matrine in all four matrices decreased with increasing doses of the C_18_ purification agent. Although C_18_ can adsorb non-polar interfering substances such as sugars, fats, lipids and sterols in the sample matrix, its adsorption of pigments was not satisfactory. As shown in Figure 4C, the recovery of quinoa roots and stems was inversely proportional to the purifier dose when using different doses of Florisil. Quinoa root, stem and leaf recoveries were 60–80%, 60–65% and 60–75%. There was no significant difference between seeds with less than 70% recovery at 30 mg Florisil, 86.3% recovery at 50 mg Florisil, or 75–80% recovery at Florisil doses greater than 50 mg. As shown in Figure 4D, the recovery of matrine in quinoa roots, stems, leaves and seeds decreased as the dose of GCB increased, using different doses of GCB. It is related to the fact that GCB has a strong adsorption feature and pigment removal ability, but it also adsorbs some pesticides. With 20 mg GCB, the recoveries of matrine in quinoa roots, stems, leaves and seeds were 81.15%, 65.47%, 67.41% and 81.37%. GCB was more effective in removing pigment compared to C_18_ because of the high pigment content in quinoa plants. To reduce the damage to the machine and the waste of resources, the experimental protocol of 20 mg GCB with a suitable dose of PSA adsorbent was used to purify the four substrates to improve their recovery.

In summary, the recovery rate and the degree of pigment purification selected the best purification method. Multiple repetitions were performed to determine the optimal ratio and dosage of the reagents used in the purification process. The highest recovery of quinoa root was 96.54% at 100 mg PSA, and the trend of decreasing recovery was observed at 150 mg, so the purification method of quinoa root was 100 mg PSA + 20 mg GCB. Quinoa stems showed the highest recovery at 100 mg PSA with 77.97%. When the dose of the purifying agent was greater than 100 mg, there was no significant increasing trend in the recovery, so its purification was 100 mg PSA + 20 mg GCB. At less than 70 mg PSA, the recovery of matrine in quinoa leaves was less than 80%. While at 70 mg, 100 mg, and 150 mg PSA, the recoveries were in the range of 80–85%, no significant differences were found between the recoveries of matrine at the three doses. So the experiment was further optimized for the dosage of PSA; 70 mg PSA + 20 mg GCB, 100 mg PSA + 20 mg GCB 150 mg PSA + 20 mg GCB were analyzed. The results showed that the recovery of matrine in quinoa leaves was ideal when 150 mg PSA + 20 mg GCB was used (Figure 4E). Less than 70 mg PSA resulted in less than 85% recovery of matrine from quinoa seeds, while 86.81%, 87.83% and 96.00% were recovered at 70 mg, 100 mg and 150 mg PSA. To reduce the waste of resources, the dosage of PSA was also further optimized and analyzed for 70 mg PSA + 20 mg GCB, 100 mg PSA + 20 mg GCB and 150 mg PSA + 20 mg GCB. The results showed that the best recovery of matrine in quinoa seeds was achieved at 70 mg PSA + 20 mg GCB (Figure 4F).

### 3.4. Evaluation of Matrix Effects

In this study, matrix effects were calculated based on the slope of the curve at six concentration levels of the matrix-matched calibration curve compared with the corresponding slope of the solvent calibration curve. A positive value of ME indicates signal enhancement of the target caused by the matrix, while a negative value indicates signal suppression. It is generally considered that when |ME| < 20%, the matrix effect is negligible, and the solvent standard curve can be used for quantitative analysis; with 20% < |ME| < 50%, the matrix effect is considered vital and a blank matrix standard curve is needed for quantitative analysis; while with an |ME| > 50%, a new pretreatment method suitable for this sample matrix needs to be established [53]. The results showed (Table 4) that the absolute values of ME of matrine in all five matrices were less than 20%. Due to the weak matrix effect, the solvent standard curve was finally used for quantitative analysis in this study.

### 3.5. Validation

The prepared matrix standard solution was determined according to the above instrumental method. The standard curve was drawn with the sample’s mass concentration (x) as the horizontal coordinate and the peak area (y) as the vertical coordinate. The linear regression equations of matrine in soil, quinoa roots, stems, leaves and seeds were y = 2,866,193x + 490,856, y = 3,214,181x + 7320, y = 3,282,544x + 7607, y = 3,042,360x + 45,875 and y = 3,056,940x + 7513 with the correlation coefficients of R^2^ of 0.9994, 0.9993, 0.9992, 0.9993 and 0.9994. It was shown that the mass concentration and peak area of matrine were linearly correlated in the range of 0.005–1 mg/L. The LOD of matrine in soil, quinoa roots and seeds was 0.001 mg/kg, and the LOQ was 0.005 mg/kg. The LOD of matrine in quinoa stems and leaves were 0.003 mg/kg, and LOQ was 0.01 mg/kg for both; details are shown in Table 4. The linear equation, LOQ and LOD followed the pesticide residue detection requirements (NY/T 788-2018) [38].

Around 2.0 g each of quinoa plant samples without matrine and 10.0 g of soil samples were weighed. The addition recovery test was performed in blank soil at three concentration levels of 0.01, 0.1 and 1 mg/kg. Matrine was set at three concentration levels of 0.1, 1.0 and 10 mg/kg in the blank quinoa root, stem, leaf and seed substrates for the recovery test. Five parallel injections were performed at each concentration level, and the determination was carried out according to the above experimental method. The recoveries were calculated. The results showed (Table 5) that the matrine in soil, quinoa roots, stems, leaves and seeds were 86.42–89.76%, 80.59–98.37%, 72.42–87.03%, 74.62–89.72% and 88.93–94.42%, respectively. The RSDs were 3.08~6.36%, 2.63~6.84%, 2.22~6.45%, 1.25~4.64% and 2.02~4.12%, respectively, which met the requirements for pesticide residue analysis. Intraday precision was determined by analyzing six parallel determinations of each spiked level over 1 day. Interday precision was calculated by analyzing three consecutive days of measurements with three replicates per day. The intra- and inter-day precision of the method was expressed as RSD [54,55]. The results showed (Table 5) that the intraday and interday precision studies of the five matrices fulfilled the requirements for pesticide residue detection with RSDs below 8.72% and 9.43%.

### 3.6. Application to Real Samples

#### 3.6.1. Dissipation Dynamics of Matrine in Quinoa Plants and Soil

By regulating the residue dissipation test protocol in the field, 0.6% matrine aqueous at 720.36 g a.i./hm^2^ was applied one, two and three times at the 4–6 leaf stage of quinoa with three replications per treatment, sampled at 0 d, 1 d, 3 d, 5 d, 7 d and 14 d. The residue dynamics details were determined by analyzing the samples from the quinoa plant and soil for the dissipation dynamics test and are shown in Table 6. Figure 5 shows that the dissipation of matrine on quinoa seeds, stems, leaves and soil showed a fast and slow trend. The dissipation dynamics of 0.6% matrine aqueous at 720.36 g a.i./hm^2^ on quinoa plants (seeds, stems, and leaves) and soil satisfied the first-order kinetic equation [56]. Residues of matrine below 0.01 mg/kg were detected on roots after one, two and three applications. The residues of matrine on soil, quinoa seeds, leaves and stems were 83.58%, 88.55%, 86.37% and 97.22% on the fifth day after the first application; the residues of matrine on soil, quinoa seeds, leaves and stems were 78.39%, 83.44%, 86.25% and 94.09% on the fifth day after the second application; the residues of matrine on soil, quinoa seeds, leaves and stems were 76.63%, 69.08% and 78.09% on the fifth day after the third application. Matrine residues were detected on soil, quinoa seeds, leaves and stems at less than 0.01 mg/kg after one, two and three applications on the fourteenth day. The residual degradation of matrine with time was fitted with a first-order kinetic equation. Results are shown in Table 7. The half-lives of matrine on soil, quinoa seeds, leaves, and stems after one application were 0.97 d, 1.28 d, 1.03 d and 0.81 d, respectively; after two applications, the half-lives of matrine on soil, quinoa seeds, leaves and stems were 0.95 d, 1.29 d, 1.04 d and 0.82 d, respectively; after three applications, the half-lives of matrine on soil, quinoa seeds, leaves and stems were 0.93 d, 1.32 d, 1.21 d and 0.94 d. The results showed that matrine is a readily degradable pesticide. The degradation rates of matrine on quinoa plants and soil, fitted according to the first level kinetic equation, were stem > soil > leaf > seed after all three applications.

#### 3.6.2. Terminal Residues of Matrine in Quinoa Plants and Soil

Based on the above sample pretreatment and instrumental conditions residue analysis methods, the final residue test samples of one, two and three applications were analyzed and determined, with the results shown in Table 8. The residues of matrine decreased significantly with an increase in the sample interval at the doses of 360.18–720.36 g a.i./hm^2^ with the application times of one, two and three times. The residues of matrine in soil were 0.104–0.774 mg/kg at the last application interval of 3 d. At the interval of 7 d, the residues of matrine in soil were 0.013–0.029 mg/kg after the third application, but the residues of matrine were lower than 0.01 mg/kg at the remaining doses and application times. The residues were lower than 0.01 mg/kg at 14 d intervals. At 3 d, 7 d, and 14 d between the last application, the residues of matrine in quinoa seeds were 0.472–1.433, <0.01–0.106, and <0.01 mg/kg; the residues of matrine in quinoa leaves were 0.332–1.018, <0.01–0.039, and <0.01 mg/kg; the residues of matrine in quinoa stems were 0.152–0.858, <0.01–0.032, <0.01 mg/kg. Results showed that the residues of matrine in quinoa plants and soil increased with increasing application dose and application frequency and gradually decreased with the extension of the harvesting period. As a plant source pesticide, the dissipation rate of matrine in soil was greatly influenced by factors such as soil physicochemical conditions and environmental conditions of the test site.

### 3.7. PHI of Matrine and Risk Assessment

To our knowledge, there are no published studies on the PHI of matrine. The deposition of matrine in quinoa seeds under three applications at 0 d was determined to be 1.574 mg/kg, 2.117 mg/kg and 3.079 mg/kg. To ensure the safe consumption of quinoa seeds, the MRL value (1 mg/kg) was based on the national standard for food safety on oranges and mandarins. Based on these results, the final residue levels of the target compounds in quinoa seeds harvested at PHI (0.84 d–2.14 d) were below the MRL established in China. These results also provide the necessary information for conducting a dietary risk assessment. The quinoa industry annual report shows that the average adult consumption is 0.1 kg/d, and GB 2763 (2021) specifies an ADI value of 0.1 mg/kg bw for matrine. This resulted in a risk index (RI) of 0.00168% for matrine in quinoa. The results showed that the dietary risk of quinoa after treatment with matrine was in the acceptable range.

### 3.8. Next Generation Sequencing to Study the Effect of Matrine on Different Soil Bacterial Communities

#### 3.8.1. Evaluation of Sequencing Depth and Sequencing Results of Soil Samples

A total of 54 samples from three soils (x for Haixi, b for Haibei, and d for Haidong) were measured in this experiment. A total of 22,234 optimized sequences were obtained by high-throughput sequencing, with an average effective read length of 250 bp. The classify-sklearn algorithm of QIIME2 was used to annotate species for each ASV using a pre-trained Naive Bayes classifier [57,58]. A total of 44 phylum, 115 class, 304 order, 501 family, 1121 genus and 714 species were obtained based on the annotated results of ASVs and the feature list of each sample. The number of randomly selected sequencing strips from a sample was used as the horizontal coordinate, and the number of species observed by the number of sequencing strips was used as the vertical coordinate to plot the dilution curve (Figure 6A), which was used to reflect the sequencing depth. Different samples were represented using different color curves. It was found that the number of ASVs increased gradually with the increasing number of sequences, and the individual curves gradually leveled off. This indicates that the amount of sequencing data is reasonable, and more data does not have a significant effect on the number of observed species. The species accumulation boxplot can be used to judge the adequacy of the sample size, with the horizontal coordinate as the sample size and the vertical coordinate as the number of ASVs after sampling. The species accumulation boxplot position tended to level off as the sample size increased (Figure 6B). Therefore, it is considered to reflect the community and structure of bacteria in each soil sample, and the sequencing depth and data volume are reasonable.

Figure 7 shows the Venn diagrams of the distribution of bacterial ASVs in soil samples at 3 d and 10 d with the application of matrine at effective doses of 0, 0.1 and 5.0 mg/kg. The total number of ASVs between the blank soil sample (x.CK.3) and the two treatment groups (x.L.3, x.H.3) at 3 d of the matrine soil treatment (Figure 7A) was 1319. The number of unique bacterial ASVs in the soil samples treated with matrine at an effective dose of 0.1 mg/kg (x.L.3) was 1792. The number of unique bacterial ASVs in soil samples treated with an effective dose of 5.0 mg/kg (x.H.3) was 1,448. At 10 d (Figure 7B), the total number of ASVs between the control (x.CK.10) and treated groups (x.L.10, x.H.10) was 1507. The number of unique bacterial ASVs in the soil samples treated with 0.1 mg/kg (x.L.10) was 1451 and the number of unique bacterial ASVs in the soil samples treated with 5.0 mg/kg (x.H.10) was 1247. The total number of ASVs between the blank soil sample (b.CK.3) and the two treatment groups (b.L.3, b.H.3) was 1551 at 3 d of the Haibei soil treatment (Figure 7C). The number of unique bacterial ASVs in the soil samples treated with matrine at an effective dose of 0.1 mg/kg (b.L.3) was 1388. The number of unique bacterial ASVs in the soil samples treated with an effective dose of 5.0 mg/kg (b.H.3) was 1283. At 10 d (Figure 7D), the total number of ASVs between the control (b.CK.10) and treated groups (b.L.10, b.H.10) was 1394. The number of unique bacterial ASVs in the soil samples treated with 0.1 mg/kg (b.L.10) was 1375, and the number of unique bacterial ASVs in the soil samples treated with 5.0 mg/kg (b.H.10) was 1679. The total number of ASVs between the blank soil sample (d.CK.3) and the two treatment groups (d.L.3, d.H.3) at 3 d of the Haidong soil treatment (Figure 7E) was 1349. The number of unique bacterial ASVs in the soil samples treated with matrine at an effective dose of 0.1 mg/kg (d.L.3) was 2387. The number of unique bacterial ASVs in the soil samples treated with 5.0 mg/kg (d.H.3) was 1880. At 10 d (Figure 7F), the total number of ASVs between the control (d.CK.10) and treated groups (d.L.10, d.H.10) was 1447. The number of unique bacterial ASVs in the soil samples treated with 0.1 mg/kg (d.L.10) was 1969, and the number of unique bacterial ASVs in the soil samples treated with 5.0 mg/kg (d.H.10) was 1540. The results showed that an increase in the number of ASVs was significantly higher after the treatment with 0.1 mg/kg of matrine than after the treatment with 5.0 mg/kg of matrine. Similarly, the bacterial abundance was higher after 0.1 mg/kg of matrine than after 5.0 mg/kg of matrine.

#### 3.8.2. Effect of Matrine on the Structure of Different Soil Bacterial Communities

Figure 8 shows the structural composition of the bacterial community between the different treatments of the three soils at the phylum taxonomic level. The main phylum was *Proteobacteria* (26.45–39.20%), *Firmicutes* (1.42–1.78%), *Actinobacteriota* (14.09–29.72%), *Acidobacteriota* (6.07–14.94%), *Bacteroidota* (4.57–9.24%), *Crenarchaeota* (1.54–6.70%), *Chloroflexi* (4.70–8.57%), *Gemmatimonadota* (3.62–8.03%) and other bacterial phyla. The most superior phylum for each treatment was *Proteobacteria,* and the second most superior phylum was *Actinobacteriota*.

The relative abundance inhibition of *Proteobacteria* was 11.56% and 12.47% at the effective dose of 0.1 mg/kg (x.L.3) and 5.0 mg/kg (x.H.3), respectively, on Haixi soil at 3 d. The relative abundance growth rate of *Firmicutes* was 81.93% and 87.50%, and that of *Bacteroidota* was 43.83% and 33.69%, respectively. At 10 d, the effective dose of 0.1 mg/kg (x.L.10) inhibited the relative abundance of *Proteobacteria* by 12.68% and −0.99% and the relative abundance of *Firmicutes* by 0.17% and 9.88%, respectively. The relative abundance inhibition rates of *Firmicutes* were 9.45% and 25.84% for the effective doses of 0.1 mg/kg (b.L.3) and 5.0 mg/kg (b.H.3) at 3 d in Haibei soil, respectively. The relative abundance growth rates of *Actinobacteriota* were 6.97% and 18.83%, respectively. At 10 d, the effective doses of 0.1 mg/kg (b.L.10) and 5.0 mg/kg (b.H.10) inhibited the relative abundance of *Firmicutes* by 22.48% and 9.51%, and *Actinobacteriota* by 29.70% and 11.12%, respectively. The relative abundance inhibition rates of the suboptimal phylum *Actinobacteriota* were 10.20% and 17.51% for the effective doses of 0.1 mg/kg (d.L.3) and 5.0 mg/kg (d.H.3) on Haidong soil at 3 d, respectively. The relative abundance growth rates of the *Firmicutes* were 18.01% and 18.47%, respectively. At 10 d, the effective doses of 0.1 mg/kg (d.L.10) and 5.0 mg/kg (d.H.10) inhibited the relative abundance growth rates of the *Actinobacteriota* by 4.52% and 14.51%, respectively. The relative abundance inhibition rates were 40.56% and 22.05% for *Gemmatimonadacea*, respectively. The results showed that the inhibitory effect on some bacterial flora increased with the increase in matrine application. Meanwhile, the inhibitory effect on bacterial flora was weakened with time.

Figure 9 shows the structural composition of the bacterial community among the different treatments of the three soils at the generic taxonomic level. The main genera were *Skermanella* (3.05–15.45%), *Lactobacillus* (0.00–3.49%), *Aminobacter* (0.15–0.78%), *Ralstonia* (0.02–2.73%), *Candidatus_Nitrocosmicus* (0.23–3.07%), *Clostridia_UCG-014* (0.00–2.00%), *Pseudarthrobacter* (1.12–4.09%), *Sphingomonas* (0.95–4.35%), *Nocardioides* (0.68–1.85%), *Bacillus* (0.32–2.69%), and a variety of unnamed bacteria. Among the bacterial genera observed and identified, the dominant species group in the soils of Haixi, Haibei and Haidong was *Skermanella*.

The relative abundance of *Aminobacter* and *Sphingomonas* in the Haixi control soil (x.CK.3) were both 0.18% and 2.50%. At 10 d (x.CK.10), their relative abundances were 3.04% and 1.18%. The effective doses of 0.1 mg/kg (x.L.3) and 5.0 mg/kg (x.H.3) reduced the relative abundance of *Skermanella* by 0.72% and 4.88% at 3 d in Haixi soil. The relative abundance of *Lactobacillus* increased by 1.61% and 3.49%. At 10 d, the effective doses of 0.1 mg/kg (x.L.10) and 5.0 mg/kg (x.H.10) reduced the relative abundance of *Skermanella* by 4.22% and 5.94%, respectively. The relative abundance of *Ralstonia* in the control Haibei soil (b.CK.3) was 0.12%. At 10 d (b.CK.10), its relative abundance was 1.15%. The relative abundance of *Lactobacillus* was reduced by 0.26% at the effective dose of 0.1 mg/kg (b.L.3) and increased by 0.29% at the effective dose of 5.0 mg/kg (b.H.3) in Haibei soil at 3 d. At 10 d, the effective doses of 0.1 mg/kg (b.L.10) and 5.0 mg/kg (b.H.10) increased the relative abundance of *Lactobacillus* by 0.92% and 0.28%. The relative abundance of *Pseudarthrobacter* in the control Haidong soil (d.CK.3) was 2.26%. At 10 d (d.CK.10), its relative abundance was 0.11%. The relative abundance of *Pseudarthrobacter* was reduced by 0.79% and 0.70% at the effective dose of 0.1 mg/kg (d.L.3) and 5.0 mg/kg (d.H.3) on Haidong soil at 3 d. At 10 d, the effective doses of 0.1 mg/kg (d.L.10) and 5.0 mg/kg (d.H.10) increased the relative abundance of *Pseudarthrobacter* by 0.23% and 1.01%, respectively. The results showed that applying a certain dose of matrine could change the relative abundance of dominant genera of soil bacteria.

## 4. Discussion

Compared to the use of solid phase extraction (SPE) [9], microwave-assisted extraction (MAE) [59], supercritical fluid extraction (SFE) [60] and molecular Imprinting Technique (MIT) [61], the QuEChERS method is fast, efficient, time-saving and simple to use, eliminating the need for complex clean-up processes as well as reducing pretreatment time and the use of organic solvents. Compared to the application of thin layer scanning [26,62], high-performance liquid chromatography [63,64] and capillary electrophoresis [65] methods for the determination of matrine bases, the LC-MS/MS method has high detection sensitivity and accuracy and low detection limits, thus, ensuring rapid and accurate analysis results of the samples. Matrine has also been found to be a readily degradable pesticide in several studies. In wheat field soil, cabbage field soil and apple orchard soil, matrine degraded quickly with a half-life of 3–6 d [18]. The degradation half-lives of matrine were 5.18–6.70 d (tomato) and 7.45–8.08 d (soil) in an open field and greenhouse cultivation [66]. The half-life of matrine in soil and tobacco leaves was 4.2–4.6 d [67]. In cucumber and soil, the half-lives of matrine were 5.19–7.42 d and 6.70–9.18 d, respectively [16]. Dissipation studies of pesticides in soil are necessary. The physicochemical properties of soil are important factors influencing their degradation. L. Kaur et al. [68] found that imazethapyr had the fastest dissipation rate in alkaline soils (pH = 8.0–8.8), followed by neutral soils (pH = 7.4) and acidic soils (pH = 5.0). The test soils for this experiment were located in the plateau region of the Qinghai Province (102°19′ E and 36°34′ N), and their soil physicochemical properties are shown in Table 1. The digestion half-life of matrine in the soil (pH = 8.16) was 0.97 d, while Qiu [9] found that in the test soil (pH = 7.91), the digestion half-life of matrine was 1.45 d. Due to its unique geographical conditions, Qinghai Province has high altitudes and alkaline soils. The pH was the main factor affecting the degradation of matrine. The inhibitory effect on some bacterial communities was enhanced with an increase in matrine application, while the inhibitory effect on bacterial communities was weakened with time. Lu et al. [69] found that microbial diversity was negatively correlated with the amount of pesticide residues in soil samples. The relative abundance of different bacteria in the soil samples varied. *Firmicutes*, *Chloroflexi*, *Actinobacteria*, *Cyanobacteria* and *Armatimonadetes* were resistant to pesticide toxicity and had the potential to use pesticide residues as nutrients for growth. In contrast, *Proteobacteria*, *Acidobacteria*, *Nitrospirae*, *Latescibacteria*, *Gemmatimonadetes*, *Verrucomicrobia* and *Chlorobi* are sensitive to pesticide residues. They have the potential to be used as indicators for assessing pesticide residue levels. Fernandes A et al. [70] found an increase in the abundance of *Enterobacteriaceae* and *Burkholderiaceae* at week 4 after the application of atrazine. The abundance of *Conexibacteraceae*, *Solirubrobacteraceae* and *Gaiellaceae* also increased at week 8. At 12 weeks after the application of atrazine, the bacterial community in the soil consisted mainly of members of the *Proteobacteria* and *Actinobacteria* families.

## 5. Conclusions

An analytical method was developed for the determination of matrine residues in quinoa plants and soil by LC-MS/MS with the QuEChERS technique for the clean-up of quinoa plants and soil. The method is simple, sensitive, accurate, reliable and widely applicable and can achieve rapid multi-residue determination of matrine to a certain extent. The half-life of matrine in quinoa seeds, leaves, stems and soil was less than 30 d. It is a readily dissipative pesticide. The final residue levels of the target compounds in quinoa seeds harvested at PHI (0.84 d–2.14 d) were all below the Chinese MRLs. The risk index RI for matrine in quinoa was 0.00057%, which is within the acceptable dietary risk range.

Next Generation Sequencing was used to identify the bacterial communities and composition of the soils at the three sites after high and low doses of matrine treatment. The bacterial communities and composition of the soils in different areas differed due to environmental factors. The bacterial community and composition also differed in the same area after treatment with high and low different doses of matrine. The results showed that the inhibitory effect on some bacterial flora groups increased with matrine application. At the same time, the inhibitory effect on bacterial groups diminished with time. The application of certain doses of matrine was able to change the relative abundance of dominant bacterial genera in soil bacteria. These findings provide a primary basis for understanding the structure and composition of soil bacterial communities in the Haixi, Haibei and Haidong areas of the Qinghai Province with the application of high and low doses of matrine.

## Figures and Tables

**Figure 1 foods-12-01337-f001:**
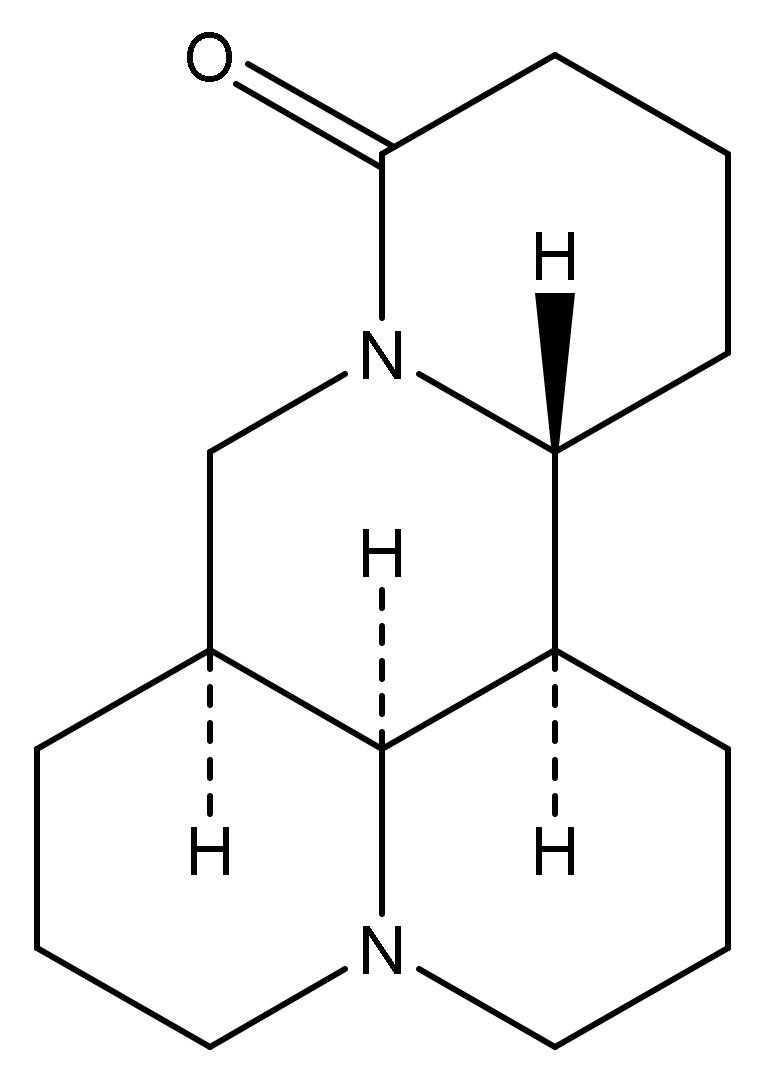
The chemical formula of matrine.

**Figure 2 foods-12-01337-f002:**
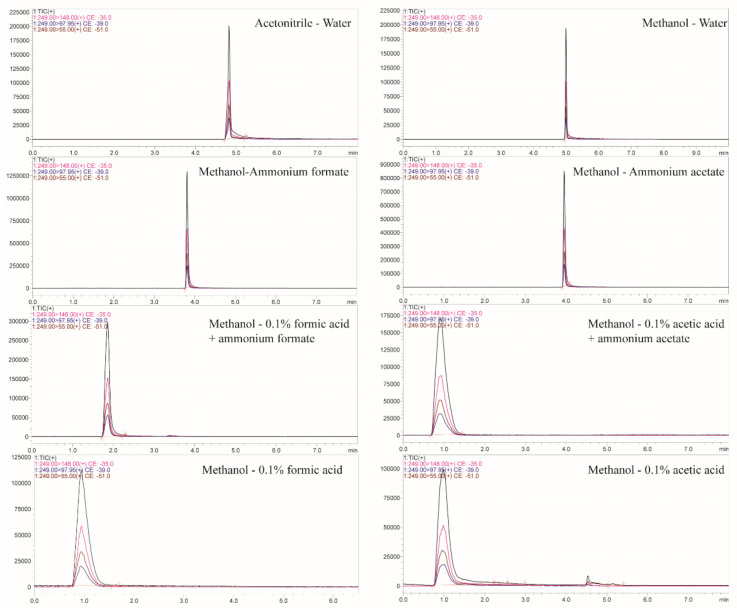
Chromatogram with different mobile phases.

**Figure 3 foods-12-01337-f003:**
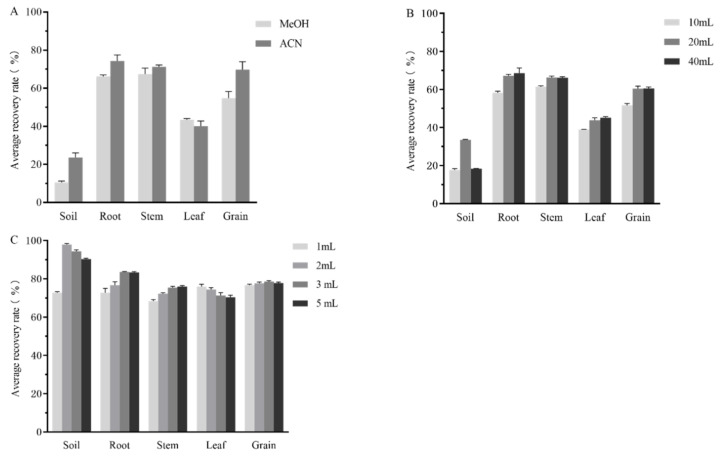
Optimization of extractants for matrine in soil and quinoa plants. (**A**) Selection of extractants for different substrates; (**B**) optimization of extractant dosage for different substrates; (**C**) optimization of ammonia dosage for different substrates.

**Figure 4 foods-12-01337-f004:**
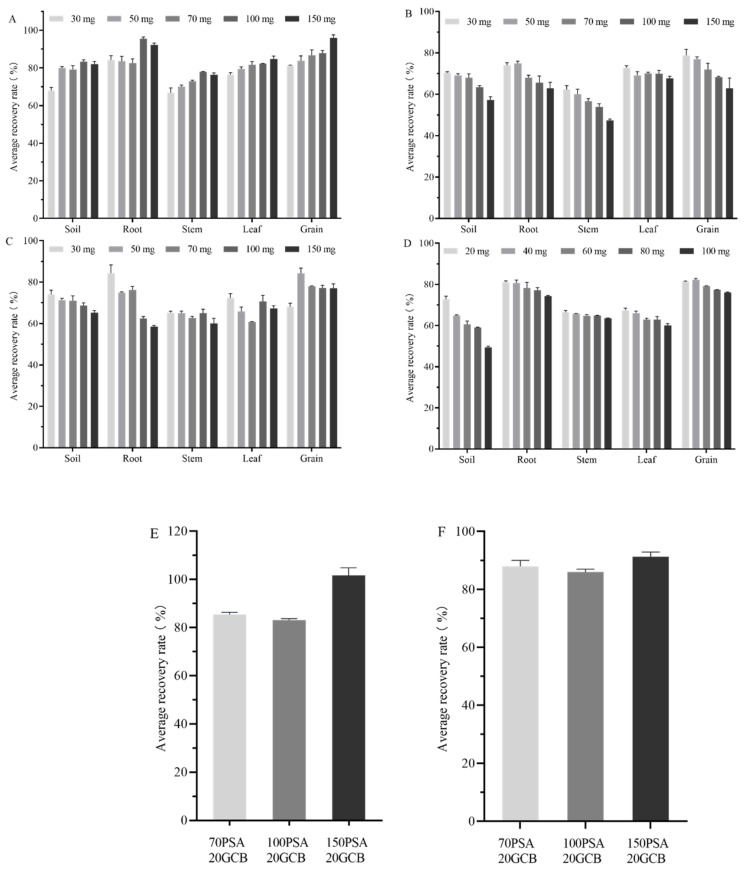
Optimization of matrine purifying agent in soil and quinoa plants. (**A**) Optimization of PSA purifier; (**B**) optimization of C_18_ purifier; (**C**) optimization of Florisil purifier; (**D**) optimization of GCB purifier; (**E**) optimization of leaf purifier; (**F**) optimization of seed purifier.

**Figure 5 foods-12-01337-f005:**
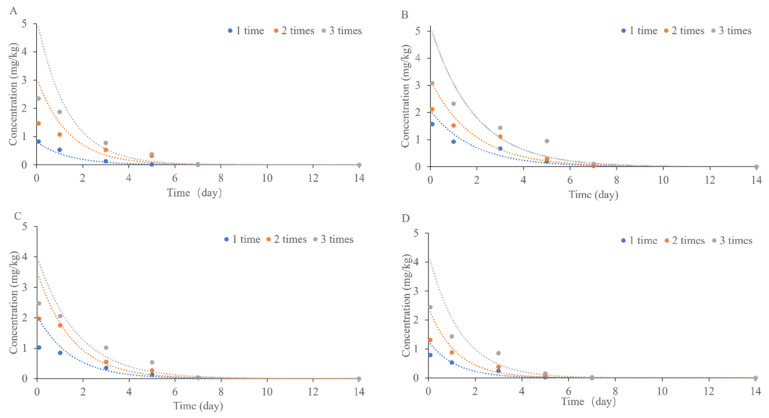
Residual dissipation curves of matrine in quinoa plants and soil under different application times (**A**–**D**) represent soil, quinoa seeds, leaves and stems, respectively.

**Figure 6 foods-12-01337-f006:**
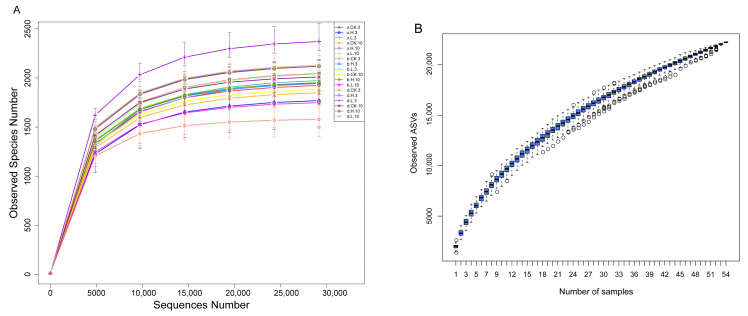
The rarefaction curves (**A**) and species accumulation boxplot (**B**) analysis.

**Figure 7 foods-12-01337-f007:**
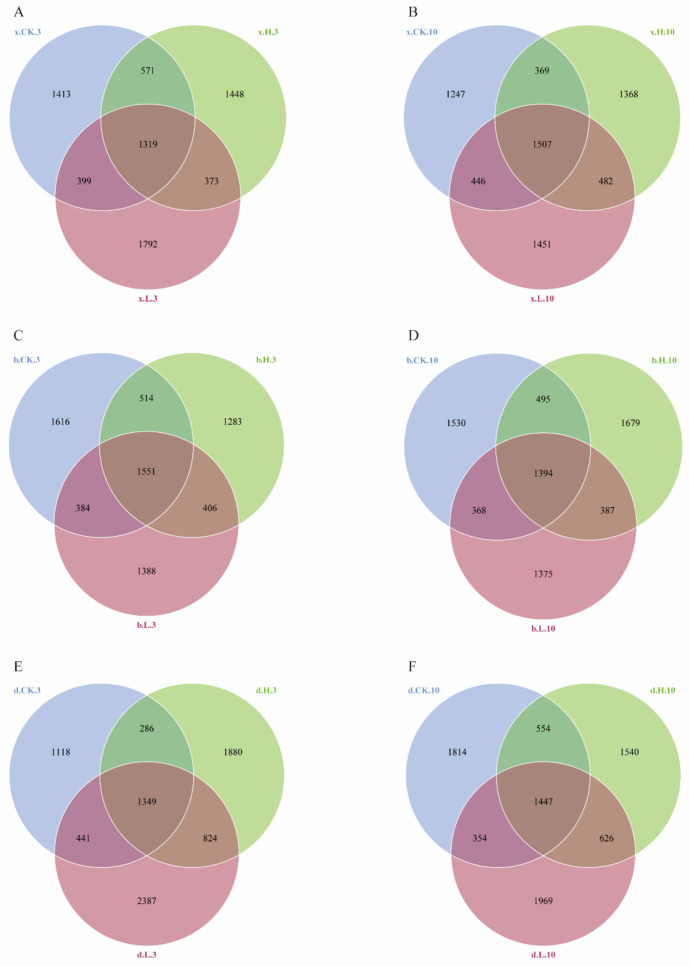
Veen diagram of the distribution of bacterial ASVs in soil samples treated with different doses of matrine. Veen diagram of the distribution of bacterial ASVs in soil samples on the third day of Haixi (**A**); Veen diagram of the distribution of bacterial ASVs in soil samples on the 10th day in Haixi (**B**); Veen diagram of the distribution of bacterial ASVs in soil samples on the 3rd day of Haibei (**C**); Veen diagram of the distribution of bacterial ASVs in soil samples on the 10th day in Haibei (**D**); Veen diagram of the distribution of bacterial ASVs in soil samples on the 3rd day of Haidong (**E**); Veen diagram of the distribution of bacterial ASVs in soil samples on the 10th day in Haidong (**F**).

**Figure 8 foods-12-01337-f008:**
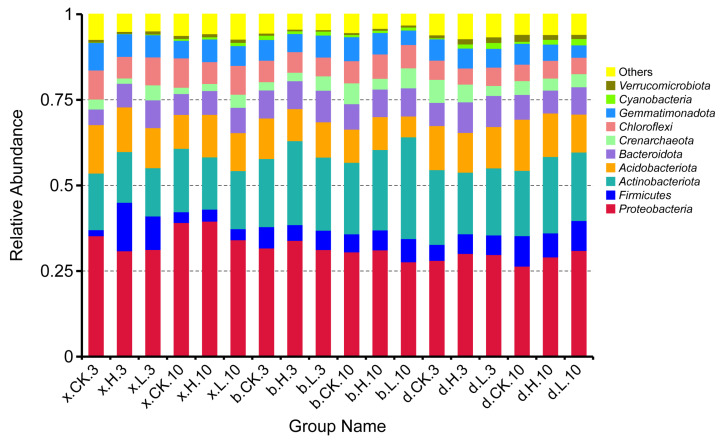
Soil bacterial phylum and its relative abundance in the three soils.

**Figure 9 foods-12-01337-f009:**
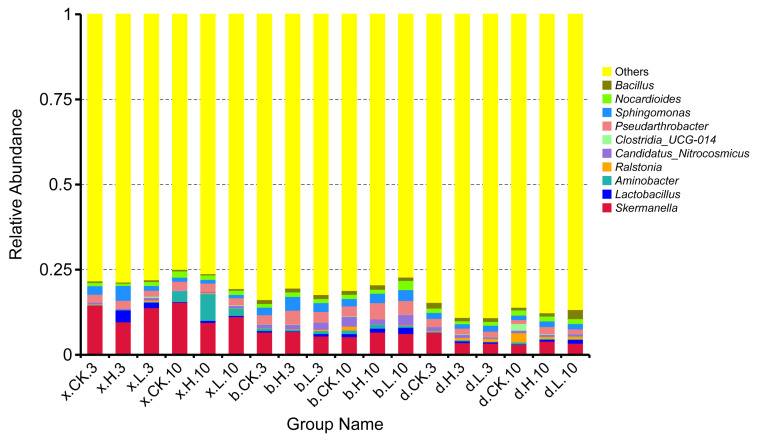
Soil bacterial genera and their relative abundance in the three soils.

**Table 1 foods-12-01337-t001:** Physico-chemical properties of the test soil.

PH	TN	TP	TK	AN	AK	AP	OM
	(g/kg)	(g/kg)	(g/kg)	(mg/kg)	(mg/kg)	(mg/kg)	(g/kg)
8.16	0.70	0.87	17.67	47.46	104	19.20	6.49

**Table 2 foods-12-01337-t002:** Parameters of mass spectra.

Ch	Precursor (*m*/*z*)	Product (*m*/*z*)	Dwell Time (msec)	Q1Pre Deviation (V)	CE	Q3Pre Deviation (V)
Ch1	249.00	148.00	100.0	−16.0	−35.0	−25.0
Ch2	249.00	97.95	100.0	−17.0	−39.0	−15.0
Ch3	249.00	55.00	100.0	−16.0	−51.0	−20.0

**Table 3 foods-12-01337-t003:** Sample numbers of different soils in the farming areas of Qinghai Province after treatment with matrine.

Soil Collection Site	Sample Time (d)	Control Group	Low Dose	High Dose
Haixi	3	x.CK.3	x.L.3	x.H.3
	10	x.CK. 10	x.L.10	x.H.10
Haibei	3	b.CK.3	b.L.3	b.H.3
	10	b.CK.10	b.L.10	b.H.10
Haidong	3	d.CK.3	d.L.3	d.H.3
	10	d.CK.10	d.L.10	d.H.10

**Table 4 foods-12-01337-t004:** The correlation equations, correlation coefficients, detection limits, quantification limits and matrix effects of matrine in five matrices.

Matrix	Linear Regression Equation	Correlation Coefficient	LOD	LOQ	Matrix Effects
		R^2^	(mg/kg)	(mg/kg)	%
Soil	y = 2,866,193x + 490,856	0.9994	0.001	0.005	0.993
Root	y = 3,214,181x + 7320	0.9993	0.001	0.005	1.120
Stem	y = 3,282,544x + 7607	0.9992	0.003	0.01	0.987
Leaf	y = 3,042,360x + 45,875	0.9993	0.003	0.01	1.003
Grain	y = 3,056,940x + 7513	0.9994	0.001	0.005	1.217

**Table 5 foods-12-01337-t005:** Validation data of the method of matrine in quinoa plants and soil.

Matrix	Supplemental Levels	Addition Recovery (*n* = 5)		Intra-Day Precision (*n* = 6)		Inter-Day Precisions (*n* = 3)	
	(mg/kg)	Rec (%)	RSD (%)	Rec (%)	RSD (%)	Rec (%)	RSD (%)
Soil	0.01	89.76	4.98	92.36	5.08	88.41	9.43
	0.1	86.42	3.08	85.13	4.61	83.37	5.65
	1	88.94	6.36	89.46	2.92	90.08	4.98
Root	0.1	98.37	2.63	85.03	6.35	83.15	7.25
	1	80.59	6.84	89.52	5.66	88.49	5.91
	10	87.81	3.72	86.08	2.28	86.58	6.35
Stem	0.1	76.07	2.22	76.49	5.41	82.43	6.58
	1	72.42	6.45	75.64	6.08	78.56	4.25
	10	87.03	2.70	81.59	4.77	81.82	7.32
Leaf	0.1	74.62	3.50	80.25	8.72	82.83	6.19
	1	86.58	4.64	73.92	7.49	79.68	5.35
	10	89.72	1.25	88.38	2.85	90.18	6.58
Grain	0.1	88.93	4.12	80.27	5.39	85.49	5.06
	1	90.96	2.02	84.35	2.47	80.51	3.61
	10	94.42	2.97	86.55	4.85	83.76	2.80

**Table 6 foods-12-01337-t006:** Dynamics of matrine dissipation in quinoa plants and soil.

Samples	Time	Application Times					
		1		2		3	
		Residue	Digestion Rate	Residue	Digestion Rate	Residue	Digestion Rate
	Day	mg/kg	%	mg/kg	%	mg/kg	%
Soil	0	0.826	0.00	1.467	0.00	2.349	0.00
	1	0.535	11.02	1.072	17.93	1.875	20.66
	3	0.125	57.65	0.534	49.97	0.774	55.54
	5	0.014	83.58	0.317	78.39	0.379	76.63
	7	<0.01	-	0.011	98.84	0.029	93.58
	14	<0.01	-	<0.01	-	<0.01	-
Grain	0	1.574	0.00	2.117	0.00	3.079	0.00
	1	0.921	37.24%	1.519	28.25	2.323	24.55
	3	0.673	57.24	1.109	47.61	1.433	53.46
	5	0.196	88.55	0.287	83.44	1.052	69.08
	7	0.036	97.71	0.065	96.93	0.106	95.56
	14	<0.01	-	<0.01	-	<0.01	-
Leaf	0	1.027	0.00	1.978	0.00	2.472	0.00
	1	0.847	17.53	1.753	11.38	2.054	16.91
	3	0.353	67.63	0.549	65.24	1.018	58.82
	5	0.14	86.37	0.172	86.25	0.535	78.36
	7	0.027	98.27	0.031	97.43	0.039	95.42
	14	<0.01	-	<0.01	-	<0.01	-
Stem	0	0.791	0.00	1.317	0.00	2.443	0.00
	1	0.536	32.24	0.883	32.95	1.438	41.14
	3	0.242	70.41	0.384	68.84	0.858	65.88
	5	0.022	97.22	0.091	94.09	0.161	92.41
	7	<0.01	-	<0.01	-	0.012	98.69
	14	<0.01	-	<0.01	-	<0.01	-

Note: the data in the table was the mean value of three times.

**Table 7 foods-12-01337-t007:** Degradation kinetic parameters of matrine in quinoa plants and soil.

Samples	Application Times	Dynamic Equation	R^2^	K	T_1/2_
Soil	1	C_t_ = 0.8172 × 10^−0.717t^	0.9849	0.717	0.97
	2	C_t_ = 3.0869 × 10^−0.732t^	0.9124	0.732	0.95
	3	C_t_ = 5.1702 × 10^−0.743t^	0.9097	0.743	0.93
Grain	1	C_t_ = 2.0671 × 10^−0.54t^	0.9484	0.540	1.28
	2	C_t_ = 3.1858 × 10^−0.537t^	0.912	0.537	1.29
	3	C_t_ = 5.175 × 10^−0.526t^	0.9142	0.526	1.32
Leaf	1	C_t_ = 2.0655 × 10^−0.676t^	0.9201	0.676	1.03
	2	C_t_ = 3.523 × 10^−0.664t^	0.9116	0.664	1.04
	3	C_t_ = 4.0129 × 10^−0.575t^	0.9216	0.575	1.21
Stem	1	C_t_ = 1.2669 × 10^−0.851t^	0.9272	0.851	0.81
	2	C_t_ = 2.4418 × 10^−0.846t^	0.9372	0.846	0.82
	3	C_t_ = 3.953 × 10^−0.751t^	0.9460	0.751	0.92

**Table 8 foods-12-01337-t008:** Terminal residues of matrine in quinoa plants and soil.

Spraying Dose	Application Times	Interval Days	Final Residues (mg/kg)			
g a.i./hm^2^		day	Soil	Grain	Leaf	Stem
360.18	1	3	0.104	0.472	0.332	0.152
		7	<0.01	<0.01	<0.01	<0.01
		14	<0.01	<0.01	<0.01	<0.01
	2	3	0.433	0.734	0.407	0.205
		7	<0.01	<0.01	<0.01	<0.01
		14	<0.01	<0.01	<0.01	<0.01
	3	3	0.635	1.104	0.849	0.582
		7	0.013	0.037	0.014	<0.01
		14	<0.01	<0.01	<0.01	<0.01
720.36	1	3	0.125	0.673	0.353	0.242
		7	<0.01	0.036	0.027	<0.01
		14	<0.01	<0.01	<0.01	<0.01
	2	3	0.534	1.109	0.549	0.384
		7	<0.01	0.065	0.031	<0.01
		14	<0.01	<0.01	<0.01	<0.01
	3	3	0.774	1.433	1.018	0.858
		7	0.029	0.106	0.039	0.032
		14	<0.01	<0.01	<0.01	<0.01

Note: The data in the table was the mean value of three times.

## Data Availability

Data is contained within this article.

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
