# Peer review of "Establishment of Residual Methods for Matrine in Quinoa Plants and Soil and the Effect on Soil Bacterial Community and Composition"

_foods, 2023, doi:10.3390/foods12061337_

Round 1

Reviewer 1 Report

The study seems to be interesting for the readers but needs careful revision

1.      L67: Sophora japonica – should be in italics

2.      Please provide more clarity on the registration status of matrine in the China. Whether the product is already approved in any crop? If yes, then there must be some information about its environmental fate and risk assessment data. Need of the study must be explained considering these points.

3.      L90-94: I would urge the author to collect the information about the method involved in fixing MRLs of matrine in different substrates. Without any analytical method, it seems impossible to estimate the residue and so as fixing the MRLs. Utility of present method needs to be explained on the basis of that.

4.      Design of experiments should be clearly mentioned.

5.      Language of materials and methods need rigorous improvement. It should not be in instructive mode.

6.      How is the solubility of matrine in methanol and acetonitrile? What is the reason behind getting better response by using methanol as solvent?

7.      Physico-chemical properties of soil need to be mentioned and correlation between dissipation and these properties, if any, needs to be explored in discussion section.

8.      More emphasis was given in citing the results. Discussion section is weak and hence, needs improvement by citing already published literature of similar work.

9.      PHI prediction and risk assessment might add value to the study. Authors may have a look on the following article and are encouraged to work out the same:  Patra, S., Das, A., Rakshit, R., Choudhury, S. R., Roy, S., Mondal, T., ... & Hossain, A. (2022). Persistence and exposure assessment of insecticide indoxacarb residues in vegetables. Frontiers in Nutrition, 9.

10.  Measurement of uncertainty shall certainly help the authors to present their data more precisely. In this regard, following article may help the team: Goon, A., Kundu, C., & Ganguly, P. (2023). Development of a Modified QuEChERS Method Coupled with LC-MS/MS for Determination of Spinetoram Residue in Soybean (Glycine max) and Cotton (Gossypium hirsutum). Journal of Xenobiotics, 13(1), 2-15.

11.   Overall, the manuscript needs language correction.

Author Response

Dear Editor,
Thank you for providing further suggestions on our manuscript. We have modified our manuscript following referee feedbacks and addressed every referee comment to the best of our knowledge. All changes were highlighted in red. Our manuscript has also been extensively edited in terms of English language/style. 
The revised manuscript is hereby submitted to be considered for publication. Thank you for your helpful coordination and understanding throughout the evaluation process of our manuscript.
All Changes Refer to the Previously Submitted Version of the Manuscript.

Please see the attachment document for more details.

Reviewer 2 Report

1. The numbering of the references will need to be done as required by Foods journal.

2.  I recommend moving lines 100-107 to the main part of the Introduction, as you have already completed the main idea with the previous paragraph. If you will be incorporating the sequencing method into your work, you will need to extend this section into the introduction.

3.   Write the purity of the solvents used (for example, Methanol, acetonitrile, acetic acid, formic acid, ammonium formate, ammonium acetate). If you have used these solvents in ratios with other solvents, you should specify in which ratio you have done so.

4.    Please make a sorting in the "Materials and methods" section. I mean start with reagents first, then write about sample preparation in full, then about equipment.

5. Lines 118-122. These lines contain information about the equipment used, but the section should only include "Chemicals and Reagents". It is recommended to make a new section on equipment, for example, "Equipment" and write there all information about the equipment. This could be combined with section 2.3.

6.     Line 127. What this unit of measure "a.i./hm" means?

7.  Line 131-132. Where these recommended ranges were taken from. Please provide a link to where the information was taken from.

8.     Line 143. What is the meaning of the word «marine»?

9.     You are studying an alkaloid called Matrine, but in Table 5 you write that you have studied pesticide residues. What did you mean when you wrote «pesticides»?

10. It will be necessary to spell out this cleaning method at the beginning of this article in the sample preparation section. This point cannot be mentioned and go straight into the discussions.

11. What is the relationship between studies of residual methods for Matrine in quinoa and the effect on soil bacterial community and composition? Why did you add research on bacteria to this manuscript?

Author Response

(The authors gave the same response as above.)

Round 2

Reviewer 1 Report

Suggestions are well-addressed.

Reviewer 2 Report

Thank you for sharing this interesting paper. The work is very well written and expertly carried out. The writing is clear and understandable, and the methods are appropriate.